# Pavlovian-To-Instrumental Transfer and Alcohol Consumption in Young Male Social Drinkers: Behavioral, Neural and Polygenic Correlates

**DOI:** 10.3390/jcm8081188

**Published:** 2019-08-08

**Authors:** Maria Garbusow, Stephan Nebe, Christian Sommer, Sören Kuitunen-Paul, Miriam Sebold, Daniel J. Schad, Eva Friedel, Ilya M. Veer, Hans-Ulrich Wittchen, Michael A. Rapp, Stephan Ripke, Henrik Walter, Quentin J. M. Huys, Florian Schlagenhauf, Michael N. Smolka, Andreas Heinz

**Affiliations:** 1Department of Psychiatry and Psychotherapy, Charité-Universitätsmedizin Berlin, 10117 Berlin, Germany; 2Department of Psychiatry and Psychotherapy, Technische Universität Dresden, 01307 Dresden, Germany; 3Neuroimaging Center, Technische Universität Dresden, 01187 Dresden, Germany; 4Zurich Center for Neuroeconomics, Department of Economics, University of Zurich, 8006 Zurich, Switzerland; 5Institute of Clinical Psychology and Psychotherapy, Technische Universität Dresden, 01187 Dresden, Germany; 6Department of Child and Adolescent Psychiatry and Psychotherapy, Faculty of Medicine, University Hospital Carl Gustav Carus, 01307 Dresden, Germany; 7Social and Preventive Medicine, Area of Excellence Cognitive Sciences, University of Potsdam, 14469 Potsdam, Germany; 8Berlin Institute of Health (BIH), 10117 Berlin, Germany; 9Department of Psychiatry and Psychotherapy, Ludwig-Maximilians-Universität München, 80336 München, Germany; 10Analytic and Translational Genetics Unit, Massachusetts General Hospital, Boston, MA 02114, USA; 11Stanley Center for Psychiatric Research, Broad Institute of MIT and Harvard, Cambridge, MA 02142, USA; 12Division of Psychiatry and Max Planck UCL Centre for Computational Psychiatry and Ageing Research, University College London, London WC1E 6BT, UK; 13Max Planck Institute for Human Cognitive and Brain Sciences, 04103 Leipzig, Germany

**Keywords:** Pavlovian-to-instrumental transfer, amygdala, alcohol, polygenic risk, high risk drinkers

## Abstract

In animals and humans, behavior can be influenced by irrelevant stimuli, a phenomenon called Pavlovian-to-instrumental transfer (PIT). In subjects with substance use disorder, PIT is even enhanced with functional activation in the nucleus accumbens (NAcc) and amygdala. While we observed enhanced behavioral and neural PIT effects in alcohol-dependent subjects, we here aimed to determine whether behavioral PIT is enhanced in young men with high-risk compared to low-risk drinking and subsequently related functional activation in an a-priori region of interest encompassing the NAcc and amygdala and related to polygenic risk for alcohol consumption. A representative sample of 18-year old men (*n* = 1937) was contacted: 445 were screened, 209 assessed: resulting in 191 valid behavioral, 139 imaging and 157 genetic datasets. None of the subjects fulfilled criteria for alcohol dependence according to the Diagnostic and Statistical Manual of Mental Disorders-IV-TextRevision (DSM-IV-TR). We measured how instrumental responding for rewards was influenced by background Pavlovian conditioned stimuli predicting action-independent rewards and losses. Behavioral PIT was enhanced in high-compared to low-risk drinkers (*b* = 0.09, *SE* = 0.03, *z* = 2.7, *p* < 0.009). Across all subjects, we observed PIT-related neural blood oxygen level-dependent (BOLD) signal in the right amygdala (*t* = 3.25, *p*_SVC_ = 0.04, *x* = 26, *y* = −6, *z* = −12), but not in NAcc. The strength of the behavioral PIT effect was positively correlated with polygenic risk for alcohol consumption (*r_s_* = 0.17, *p* = 0.032). We conclude that behavioral PIT and polygenic risk for alcohol consumption might be a biomarker for a subclinical phenotype of risky alcohol consumption, even if no drug-related stimulus is present. The association between behavioral PIT effects and the amygdala might point to habitual processes related to out PIT task. In non-dependent young social drinkers, the amygdala rather than the NAcc is activated during PIT; possible different involvement in association with disease trajectory should be investigated in future studies.

## 1. Introduction

Problematic alcohol drinking patterns like bingeing or heavy drinking during adolescence and early adulthood are associated with severe psychological, social and health problems [1]. Therefore, elucidating mechanisms that underlie high-risk drinking in young adulthood is important. Here, we assess biological factors in relation to a behavioral phenomenon that has been associated with chronic alcohol consumption theoretically [2,3,4] and empirically [5,6]. Specifically, we focus on behavioral effects of Pavlovian-to-instrumental transfer and at risk alcohol consumption in young male social drinkers, neural correlations and the association to polygenic risk for alcohol consumption.

Alcohol intake has been shown to be promoted by positive and negative contexts [7,8]. One mechanism implicated in the influence of contexts on ongoing behavior is Pavlovian-to-instrumental transfer (PIT). In general PIT, appetitive Pavlovian cues promote instrumental responses while aversive Pavlovian cues reduce such responses or even promote withdrawal independent of reward types [9]. In specific PIT, Pavlovian cues promote instrumental behavior associated specifically with the same outcome [10]. In animal models of addiction, drug exposure increases general and specific behavioral PIT effects [11,12] and enhanced food-related behavioral PIT was predictive for subsequent stronger cue-induced reinstatement of alcohol seeking [13]. We have recently reported increased nondrug-related behavioral PIT in detoxified alcohol-dependent patients compared to age-and gender-matched social drinkers using monetary cues [5]. In this study, we ask whether similar differences in PIT are measurable in an independent and much younger cohort of male high-versus low-risk social drinkers. Previous studies have examined alcohol-specific behavioral PIT effects in social drinkers but did not assess the association between behavioral PIT effects and individual drinking patterns [14], nor did they find an association with subclinical alcohol dependence [15,16] or neural PIT correlates using electroencephalography (EEG) [16]. In contrast to these studies, we investigate nondrug-related PIT effects in young high-versus low-risk [17] social drinkers on a behavioral and neural level using functional magnetic resonance imaging (fMRI). 

On a neural level, animal studies showed that the amygdala is a core region associated with behavioral PIT [10,18,19,20]. Moreover, the strength of behavioral PIT is positively correlated with dopaminergic neurotransmission in the ventral striatum [21], which in turn is known to be modulated by alcohol intake [22,23,24]. In humans, both the nucleus accumbens (NAcc) and amygdala are activated during PIT [25,26,27], and amygdala activation by alcohol cues has been positively correlated with craving in alcohol-dependent patients during an alcohol-approach bias task [28]. Interestingly, PIT-related activation of the NAcc, but not the amygdala, predicted relapse after detoxification in alcohol-dependent patients [5].

Many genes can be involved in phenotypes such as alcohol use with respectively small effect sizes [29]. Therefore, we used a polygenic risk approach to investigate the genetic influence on alcohol consumption and behavioral PIT in our sample. It has been shown that higher polygenic predisposition for alcohol problems predicts earlier initial alcohol consumption and early heavy drinking patterns, as well as more alcohol-related problems in independent samples [30,31,32]. We therefore aimed to investigate how a polygenic risk score (PRS) for alcohol consumption derived from an independent large genome-wide association study [33] is associated with alcohol consumption and behavioral PIT in our sample.

As we previously observed stronger nondrug-related behavioral PIT in alcohol-dependent patients compared to controls as well as a stronger PIT-related NAcc activation predicting relapse [5], we wanted to assess whether there are comparable differences in nondrug-related behavioral PIT between the two groups of young male high-versus low-risk drinkers. Therefore, we examined a non-clinical sample of young males and hypothesized (1) stronger nondrug-related behavioral PIT effects in high-compared to low-risk drinkers [17]; (2) PIT-related blood oxygen level-dependent (BOLD) activity in an a-priori region of interest (ROI) encompassing amygdala and NAcc; and (3) a positive association between alcohol-related polygenic risk [33] and both the strength of nondrug-related behavioral PIT and alcohol consumption in our sample.

## 2. Materials and Methods

### 2.1. Participants and Procedure

1974 males were randomly drawn from local registration offices in two sites (Berlin & Dresden, Germany [34]) shortly after their 18th birthday representing their local legal adult age. We screened 445 respondents via telephone. Exclusion criteria were left-handedness, history of major neurological or psychiatric disorders (except for nicotine dependence and alcohol abuse), current alcohol abstinence and MRI-specific contraindications. In total, 209 subjects were included and tested. After quality control, 191 behavioral, 139 imaging and 157 genetic datasets could be analyzed (see Figure 1). Subjects were descriptively comparable to similar cohorts drawn from the German general population (see Appendix A).

All participants were assessed with the Composite International Diagnostic Interview (CIDI) [35,36] according to the Diagnostic and Statistical Manual of Mental Disorders (DSM-IV-TR) [37] and completed a neuropsychological test battery. On a second appointment (mean = 8.5 (SD = 16.2) days later), participants performed a task battery during a functional magnetic resonance imaging (fMRI) scan. The experimental procedure comprised a two-step Markov decision making task [38,39] and the PIT task with nondrug-and drug-related contexts [40]. Blood samples for genetic analyses were taken at first (Berlin) or second (Dresden, after MRI scan) appointment. The study procedures (clinical trials identifier: NCT01744834) adhered to the Declaration of Helsinki and were approved by local ethics committees of Charité Universitätsmedizin Berlin (EA/1/157/11) and Technische Universität Dresden (EK 227062011). All participants gave written informed consent prior to participation.

### 2.2. Experimental Design

The PIT task consisted of four parts: 

Instrumental training. Participants collected shells by repeated button presses receiving probabilistic feedback (see Figure 2A). To control for instrumental performance, participants trained until they reached a criterion of 80% correct choices over 16 trials (for a minimum of 60 or a maximum of 120 trials).

Pavlovian conditioning. Trials began with presenting for 3 s a compound stimulus consisting of fractal-like pictures and pure tones (conditioned stimulus, CS); followed by a 3 s delay, and finally an unconditioned stimulus (US: picture of a coin) for 3 s (see Figure 2B). Participants were instructed to memorize the pairings. All participants completed 80 trials. 

Pavlovian-to-instrumental transfer. Participants performed the instrumental task now with CS tiling the background (see Figure 2C). Note that the instrumental task was independent of the value of the background stimulus. No outcomes were presented, but participants were instructed that their choices still counted towards the final monetary outcome. The pairings of CS in background und shell in foreground were counterbalanced with each combination showing three times, resulting in 90 trials. 

Forced choice task. Finally, participants chose one of two CSs (Figure 2D). All possible CS pairings were presented three times in randomized order. 

### 2.3. Self-Reported Questionnaires

We used self-reported measures for sample description measuring alcohol dependence severity (ADS) [41], alcohol craving (Obsessive Compulsive Drinking Scale, OCDS) [42] and nicotine dependence severity (Fagerström Test for Nicotine Dependence, FTND) [43]. 

### 2.4. Measures of Alcohol Consumption

In accordance with previous analyses of nondrug-related PIT group differences between alcohol-dependent patients and matched social drinkers [5], we used the World Health Organization (WHO) definition for risk of acute alcohol-related problems [17] based on average ethanol consumption on a drinking occasion within the last year. Accordingly, subjects qualified as low-risk drinkers (≤60 g of alcohol on a single occasion) or high-risk drinkers (>60 g), respectively. To further characterize participants’ drinking behavior and how this relates to polygenic risk, we calculated a sum score of drinking variables (henceforth referred to as drink score) from the z-scaled CIDI items with higher values indicating higher or more risky alcohol consumption [44] (see Appendix A for calculations ”measures of alcohol consumption”).

### 2.5. MRI Data Acquisition

Functional imaging was performed on a Siemens Trio 3 Tesla MRI scanner with an Echo Planar Imaging (EPI) sequences (repetition time, 2410 ms; echo time, 25 ms; flip angle, 80°; field of view, 192 × 192 mm^2^; voxel size, 3 × 3 × 2 mm^3^, 1 mm gap; 480 volumes) comprising 42 slices acquired in descending order and rotated approximately −25° to the bicommissural plane. For coregistration and normalization during pre-processing, a three-dimensional magnetization-prepared rapid gradient echo image was acquired (repetition time, 1900 ms; echo time, 2.52 ms; flip angle, 9°; field of view, 256 × 256 mm^2^; 192 sagittal slices; voxel size, 1 × 1 × 1 mm^3^). A field map was recorded to account for individual homogeneity differences of the magnetic field. 

The PIT task was programmed using Matlab with the Psychophysics Toolbox Version 3 (PTB-3) extension [45]. Responses during PIT were made using a current-design MRI-compatible response box with the right index finger.

### 2.6. Polygenic Risk Score

To genotype our sample, DNA was extracted semi-automatically with a Chemagen Magnetic Separation Module (Perkin Elmer) from whole blood drawn in EDTA tubes before fMRI assessment. All samples were genotyped with the Illumina Infinium Psych Array Bead Chip [46]. Content for the PsychArray includes 265,000 proven tag SNPs found on the HumanCoreBeadChip, 245,000 markers from the Human Exome Bead Chip and 50,000 additional markers.

For calculating the polygenic risk score (PRS), we used a standard approach [47]. Our training data set derived from a large genome-wide association study (GWAS) investigating the genetic basis of alcohol consumption in *n* > 105,000 healthy social drinkers [33]. To calculate a polygenic risk score for each individual in our independent sample we summed up the number of alleles for each single nucleotide polymorphism (SNP) weighted by the effect size (association between each SNP and alcohol consumption) drawn from GWAS from the training data set. The score was computed at different *p*-value thresholds (*p* = 1, *p*= 0.5, *p* = 0.2, *p* = 0.1, *p* = 0.05, *p* = 0.01) representing the composite additive effect of all SNPs (*p* = 1, *n* = 100,000 SNPs) or the number of SNPs above the respective threshold. This gives the SNPs with higher significance automatically more weight than SNPs with lower significance. 

### 2.7. Statistical Analysis

Data were analyzed in Matlab 2011a [48] and the R System for Statistical Computing Version 3.3.3 [49]. Functional magnetic resonance imaging (fMRI) data were analyzed using Statistical Parametric Mapping (SPM 12) software package [50]. All analyses refer to the transfer part of the PIT task (Figure 2C). 

### 2.8. Behavioral Analysis

We conducted a generalized linear mixed-effects model implemented in the lme4 package (version 1.1-12). In order to assess the individual contribution of Pavlovian values on behavior, we built a Poisson distributed model where the number of button presses in each trial was predicted by the value of the background CS (−2, −1, 0, +1, +2; linear effect) and the instrumental condition (collect/not collect; coded as +0.5/−0.5). The within-subject factors (intercept, main effect of CS value, instrumental condition, and their interaction) were taken as random effects across subjects. Instrumental stimuli (shells) and Pavlovian CSs were taken as separate crossed random effects with varying intercepts in order to control for potential item effects. We included group (high-versus low-risk drinkers; coded as +0.5/−0.5) as between-subject factor to this model, performing two-tailed statistical tests on the a-priori hypothesis that behavioral PIT effects are stronger in high-compared to low-risk drinkers. Furthermore, we extracted individual regression slopes from the original generalized linear mixed-effects model as a measure of individual strength of behavioral PIT for further testing the association between the strength of the behavioral PIT effect and polygenic risk for alcohol consumption. 

### 2.9. Imaging Analysis

Preprocessing. For preprocessing information see Appendix A.

First-level analysis. The influence of Pavlovian stimulus values on instrumental responses (PIT effect) was measured by constructing a linear contrast, which weighted the parametric modulator of each condition (i.e., trial-by-trial number of button presses) by their associated Pavlovian values (−2, −1, 0, +1, +2) [5], i.e., the neural PIT effect was modeled by number of button presses times value of background stimulus (onset: appearance of shell in foreground). To account for variance caused by motor responses, button presses for all trials together were modeled with a regressor of no interest. Regressors were then convolved with the canonical hemodynamic response function. The six realignment parameters and their first derivatives were included as regressors of no interest. For a measure of the neural PIT effect a linear contrast was constructed, which weighted the parametric modulators for each condition by the related Pavlovian background value. The neural CS value effect was measured with a similar linear contrast on the CS event regressors. 

Second-level analysis. Linear contrast images for neural PIT and neural CS contrasts were taken to the second level. To test for the neural PIT effect, we conducted a one-sample *t*-test. Study site was included as covariate. Using the wake Forest University (WFU) Pick Atlas software [51], we computed one ROI for a small volume correction (SVC) approach including both the bilateral NAcc and bilateral amygdala to avoid multiple testing. Next, we extracted individual mean beta values of the observed neural PIT effect to test the association of neural and individual behavioral PIT effect. We expected a positive association, yet conducted a two-tailed test. 

### 2.10. Polygenic Analyses

We computed a PRS (see methods above), to verify genetic risk for alcohol consumption computed at threshold *p* = 1, thus including all genetic signal. To present the full picture, we also report results at other *p*-levels. Spearman’s correlation coefficient was used to compute the respective association between PRS and (i) the continuous composite drink score, and (ii) behavioral PIT slope extracted from the glmer model described above. We expected a positive association between these measures and tested two-tailed. While the first analysis (i) provides evidence of whether the PRS is associated with drinking in our sample (replications see [30,31]), the second analysis (ii) explores a direct association between PRS and behavioral PIT (*p*-values for descriptive reasons only).

## 3. Results

### 3.1. Sample Characteristics by Drinking Group

Appendix A summarizes sample characteristics comparing high-risk drinkers (*n* = 94) to low-risk drinkers (*n* = 97) according to WHO stratification [17]. Pure alcohol consumed in life in kg, ADS and OCDS are for clinical description of severity of alcohol use problems. According to that, high-risk drinkers reported higher lifetime alcohol intake, stronger craving in the past seven days and more problems associated with alcohol dependence. Groups did not differ significantly in terms of smoking severity, age, socio economic status and verbal intelligence. 

### 3.2. Behavioral Results

Behavioral PIT effects were significantly stronger in high-compared to low-risk drinkers (for PIT effect in whole sample see Appendix A). Specifically, the regression analyses showed an interaction effect between Pavlovian background and group on instrumental response rate (*b* = 0.09, *SE* = 0.03, *z* = 2.7, *p* < 0.009, *n* = 191, two-tailed; see Figure 3 and Appendix A) in the way that with higher value of the background stimulus the instrumental response rate increases. Crucially, this was not due to smoking severity (see Appendix A), or differences in instrumental performance (*p* = 0.54, see Appendix A). 

### 3.3. Imaging Results

The ROI analysis (encompassing bilateral amygdalae and NAcc) for the whole sample revealed a significant PIT-related activation in the right amygdala (*t*_(137)_ = 3.25, *p*_SVC_ = 0.04, *x* = 26, *y* = −6, *z* = −12, *k* = 29, *n* = 139, see Figure 4A), which could not be explained by a pure CS effect (see Appendix A). Extracted mean beta-values within the right amygdala showed a positive association with the behavioral PIT effect (*b* = 0.07, *SE* = 0.014, *z* = 4.7, *p* < 0.001, two-tailed, *n* = 139). High-versus low-risk drinkers according to the WHO did not differ significantly in neural activation during PIT. Within our single ROI, there was no significant activation in the NAcc. For exploratory whole brain analyses at *p_uncorr_* < 0.001 and *k* = 10 see Appendix A.

### 3.4. Polygenic Risk in Association with Behavioral PIT

We found a significant positive correlation between the PRS and the composite drink score in our sample (*r_s_* = 0.17, *p* = 0.032, *n* = 157, two-tailed see Figure 5A). Figure 5B illustrates this association between polygenic risk and drink score in our sample using PRS computed at different thresholds ranging from *p* = 0.01 to *p* = 1. Furthermore, there was a significant positive correlation between the strength of the behavioral PIT effect and the PRS (*r_s_* = 0.17, *p* = 0.032, *n* = 157, two-tailed, see Figure 5C). Figure 5D illustrates this association between polygenic risk and behavioral PIT using PRS computed at different thresholds ranging from *p* = 0.01 to *p* = 1. For a multi-level approach using multi-modal information (behavioral PIT, neural PIT effect and PRS), see Appendix A.

## 4. Discussion

We aimed to investigate the nondrug-related behavioral PIT effect in a cohort of young male high-versus low-risk [17] drinkers and the neural correlate of this PIT effect. We further explored the association of behavioral PIT with polygenic risk for alcohol consumption. Our main finding of enhanced behavioral PIT in high-risk drinkers suggests that strong effects of Pavlovian cues on instrumental behavior could be a core behavioral signature of risky alcohol consumption. We further observed PIT-related amygdala activation and a positive association between the strength of the behavioral PIT effect and polygenic risk related to alcohol consumption. A similarly assessed behavioral PIT effect was previously enhanced in alcohol-dependent patients compared to matched controls [5]. However, whether this is a vulnerability marker for developing alcohol use disorder or a consequence of substance exposure requests investigation in future longitudinal studies.

On a neurobiological level, we observed PIT-related activation in the right amygdala, which is in line with animal [52] and human studies [25,26,53], supporting previous reports of a key role for the amygdala in PIT [10]. Lesions of the basolateral amygdala abolished the selective excitatory effects of reward-related Pavlovian stimuli, while lesions of the central amygdala abolish general motivational effects of such cues [10]. A human study by Prevost et al. [26] confirmed the dissociation within the amygdala and general and specific PIT. Although our task has repeatedly identified Pavlovian modulation of instrumental responses [5,6,9], we cannot distinguish between general and specific PIT effects, as the reward used for Pavlovian and instrumental conditioning was both monetary, preventing conclusions about the specificity of the Pavlovian influence on instrumental behavior. Moreover, the amygdala was involved in cue-elicited habitual rather than goal-directed reward seeking in healthy humans [54]. Thus, our results on PIT effects in the amygdala might point to habitual processes that are related to enhanced PIT effects in our task.

Interestingly, PIT-related NAcc activity was related to poor treatment outcome in alcohol-dependent patients [5] supporting the known role of chronic alcohol intake on striatal dopaminergic neurotransmission [55], while in this sample of young at-risk drinkers, we could not observe PIT-related NAcc activity. This may suggest that the neurobiology by which CSs come to guide behavior is different in at risk alcohol consumption versus alcohol-dependent patients. The amygdala may be involved in cue-induced modulation of instrumental behavior among young adults and the NAcc might be associated with the nondrug-related PIT effect in alcohol dependence only. These biological differences may be associated with the shift from impulsive to habitual drug intake during the development of substance use disorders [3,56,57,58]. Further studies should assess whether these PIT activation differences are correlated with loss of control over alcohol intake.

Moreover, we observed alcohol-related polygenic risk to be associated with higher alcohol consumption in our sample. This effect is in line with numerous studies that emphasize a polygenic risk for risky alcohol consumption [30,31]. In addition, our study reveals that this polygenic risk is also related to a possible underlying mechanism associated with risky alcohol consumption, namely PIT. Thus, the strength of contextual stimuli in influencing ongoing instrumental behavior might be modulated by an underlying genotype for alcohol consumption, strengthening PIT to be a relevant mechanism to understand alcohol intake.

In terms of clinical implications, Ostafin et al. [59] showed that prevention programs boosting the explicit motivation to reduce alcohol consumption are only effective in hazardous drinkers with low automatic approach tendencies towards alcohol, and can even result in increased alcohol consumption if they fail to address implicit motivational processes [60]. Our data suggest that high-risk drinkers may be more susceptible to cue-triggered processes; this group may particularly profit from interventions focusing on implicit motivational processes to reduce hazardous alcohol consumption [61]. 

Limitations of our study include potential selection bias during recruitment: due to ethical guidelines, we were required to state in the invitation letter that we wish to recruit for a study on alcohol consumption. Subjects with high alcohol consumption might have been more reluctant to participate. Moreover, we excluded alcohol-dependent subjects, as we aimed to investigate PIT in social drinkers only. These two reasons might explain why most of our sample reported low life-time alcohol consumption (kg pure alcohol), a comparably small number of dependence problems as measured by the ADS as well as little numbers of DSM-IV-TR alcohol abuse diagnoses. Moreover, we focused on high risk for acute alcohol-related problems as defined by the WHO. Therefore, our data cannot make conclusions about chronic alcohol problems. In line with DSM 5, a dimensional approach to AUD has been applied, which needs to also identify risky variants at the lower end which might convert into high risk drinkers. Here, the reported study design is cross-sectional, which limits conclusions about how PIT effects change over the course of alcohol consumption and it limits mechanistic statements and explanations. Moreover, we included male subjects exclusively, thus limiting the generalization to female alcohol consumers. However, female drinking has been reported to be manifested in a different way that lack some of the very public and overt ways that male drinking presents [62]. We only assessed males to avoid loss of power due to potential gender differences; findings need to be replicated for women. Furthermore, as expected, especially the genetic effects sizes are rather small, which limits the predictive power on an individual level. Finally, we cannot draw conclusions about other substance-related disorders, as we focused our analyses on alcohol only.

In summary, we observed enhanced behavioral PIT effects in young high-risk drinkers, which were associated with functional activation in the right amygdala and correlated with an alcohol-related polygenic risk. Together with previous findings on behavioral PIT effects in alcohol-dependent patients [5], these data suggest that stronger behavioral PIT effects are a trait marker for high-risk alcohol use. How this is related to the risk of developing alcohol dependence should be further explored in longitudinal studies. Although the behavioral effects reported here were similar to the previously reported effects in patients, the neural correlates involved the amygdala rather than the NAcc [5], suggesting a differential involvement of these structures at different time points in the disease trajectory. Given the well-documented effects of alcohol on striatal dopaminergic neurotransmission [55], future studies should explore alcohol effects on striatal versus amygdala function and their malleability in longitudinal settings.

## Figures and Tables

**Figure 1 jcm-08-01188-f001:**
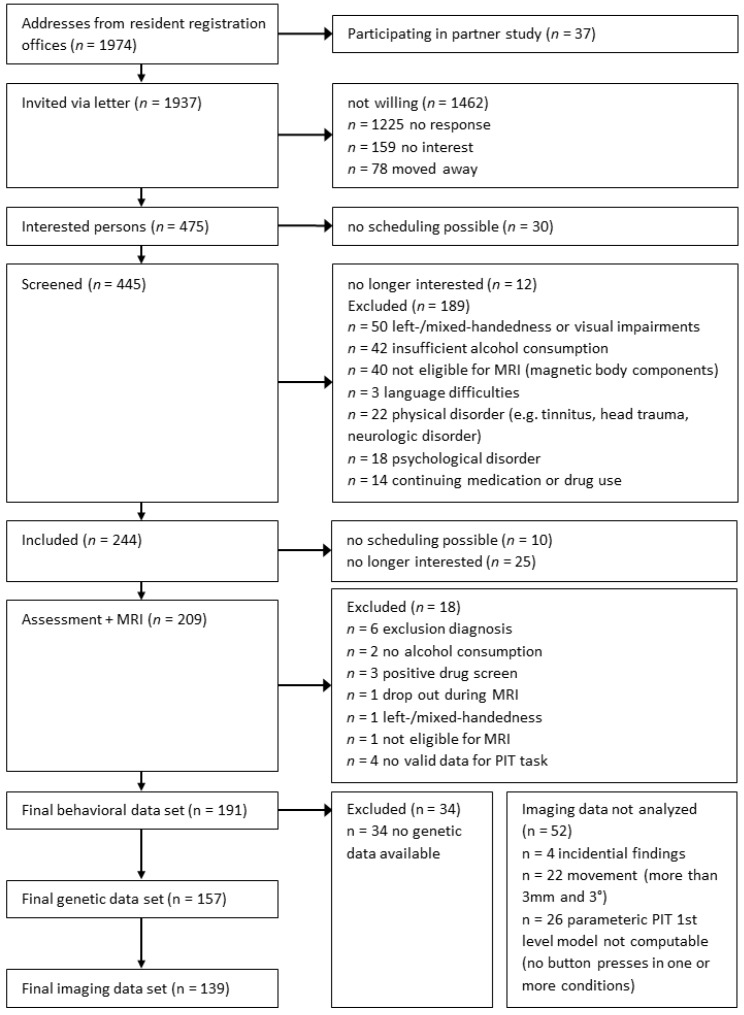
Recruiting and exclusion procedure leading to the final behavioral, genetic and imaging datasets. MRI: magnetic resonance imaging; PIT: Pavlovian-to-instrumental transfer.

**Figure 2 jcm-08-01188-f002:**
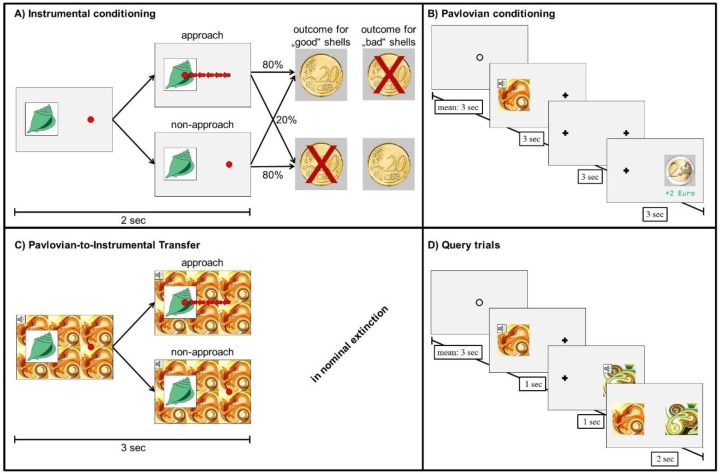
Pavlovian-to-instrumental Transfer (PIT) task. (**A**): Instrumental training: collecting a ’good’ shell was rewarded in 80% while not collecting a ‘good’ shell punished in 80%. The opposite reinforcement contingencies applied to ’bad’ shells. Red arrows indicate the five or more button presses required to approach and collect the presented shell. By trial and error, subjects learned to collect or not to collect three out of six shells. (**B**): Pavlovian conditioning: subjects passively viewed a conditioned stimulus (CS), which was deterministically followed by an unconditioned stimulus (US). As CS, a compound of a tone and five fractal-like visual stimulus was used. USs were pictures of a coin (−2€, −1€, 0€, +1€, +2€). (**C**): Transfer: subjects were asked for the instrumental response, while the background was tiled with the CS. Trials with drink-related background stimuli are not displayed. (**D**): Query trials: Subjects were asked to choose the better (i.e., that was associated with the highest reward or lowest punishment during Pavlovian conditioning) between sequentially presented CSs.

**Figure 3 jcm-08-01188-f003:**
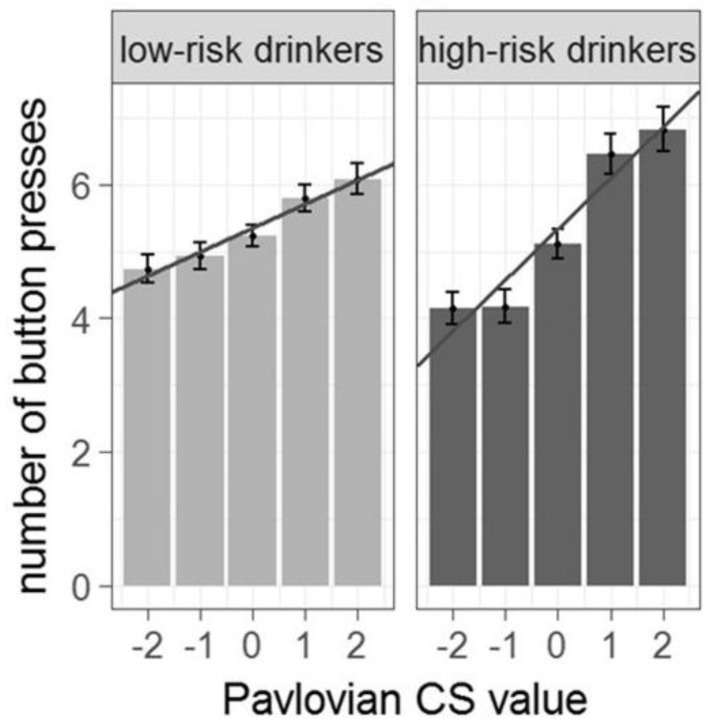
Behavioral PIT effect in low-versus high-risk drinkers (*n* = 191). Number of button presses for each Pavlovian background condition. The behavioral PIT effect is stronger in high-risk drinkers (as indicated by a steeper group regression slope).

**Figure 4 jcm-08-01188-f004:**
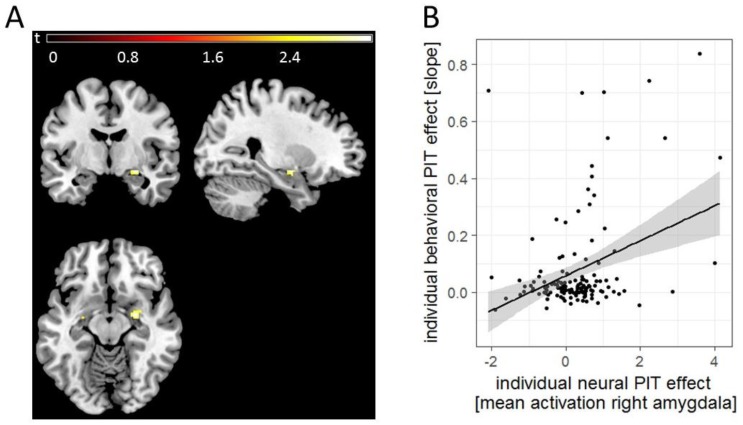
(**A**). Neural PIT effect in the right amygdala for the whole group (*n* = 139). For illustrational purposes, this effect was masked for the bilateral amygdala (region of interest (ROI) derived from wake Forest University (WFU) Pick Atlas). (**B**). The PIT-related activation in the right amygdala positively correlated with the behavioral PIT effect.

**Figure 5 jcm-08-01188-f005:**
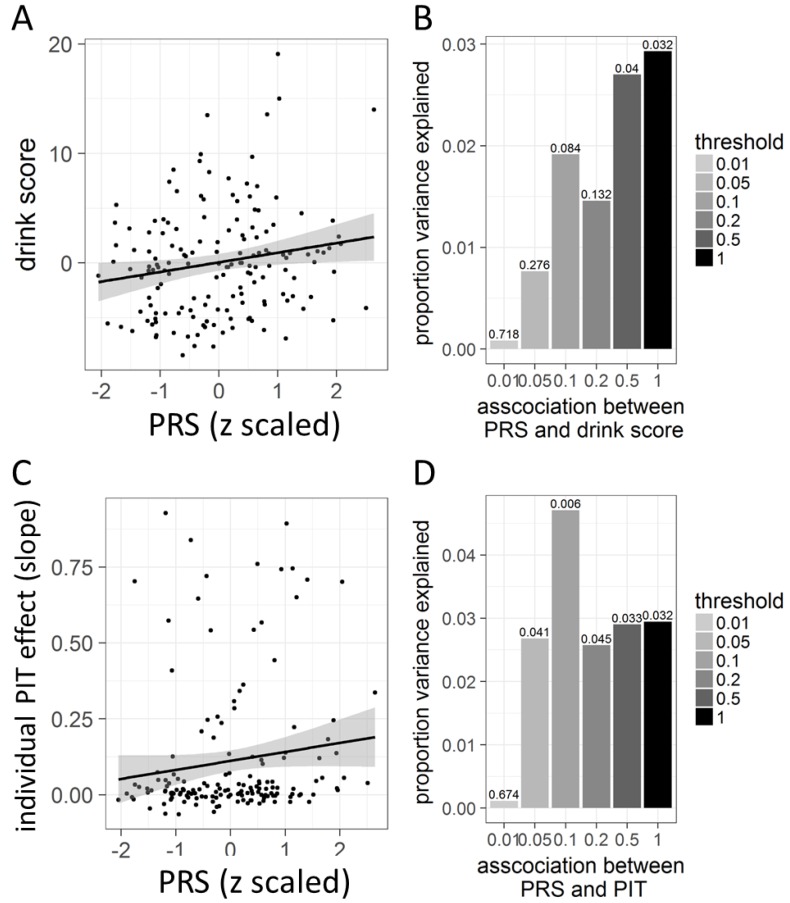
Polygenic risk score (PRS) for alcohol consumption in association with alcohol intake and PIT in our sample (*n* = 157). (**A**): Association between PRS and alcohol intake as measured by the drink score in our sample. (**B**): Explained variance of the association between PRS and drink score as indicated by r_s_^2^ for each threshold. Values at each bar represent the p-values, tested two-tailed. (**C**): Association between PRS and behavioral PIT effect slope. (**D**): Explained variance of the association between PRS and behavioral PIT slope as indicated by r_s_^2^ for each threshold. Values at each bar represent the *p*-values, tested two-tailed.

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
