# Peer review of "Pavlovian-To-Instrumental Transfer and Alcohol Consumption in Young Male Social Drinkers: Behavioral, Neural and Polygenic Correlates"

_jcm, 2019, doi:10.3390/jcm8081188_

Round 1

Reviewer 1 Report

The authors aimed to investigate non-drug-related Pavlovian-to-instrumental transfer (PIT) in a sample of young, male high- and low-risk social drinkers. They assessed behavioral PIT effects, BOLD activation in a priori defined regions of interest, namely, amygdala and nucleus accumbens, and associations with alcohol consumption, as well as a polygenic risk score for alcohol consumption. Results showed that behavioral PIT was increased in high-risk compared to low-risk drinkers. Across groups, there was a significant PIT-related BOLD activation in the right amygdala but not in the nucleus accumbens. There was a positive correlation between the behavioral PIT effect and the polygenic risk score for alcohol consumption. 

The present manuscript is well written. The rationale and methodology are clearly described, and the conclusions are justified by the data. Suggestions to improve the manuscript are listed below.

·      Abstract line 42: The authors should be precise with regard to which PIT effect they are referring to (behavioral or neural). 

·      Abstract, line 43ff: The authors conclude that PIT-related amygdala activation might be a biomarker for a subclinical phenotype of risky alcohol consumption, however, such an association between PIT-related amygdala activation and the polygenic risk score or reported alcohol consumption is not reported in the manuscript. Was PIT-related amygdala BOLD activation associated with the polygenic risk score or alcohol consumption? If so, this should be reported; if not, this should still be reported, discussed, and reflected in the last sentence of the abstract.

·      Methods, 2.10. Polygenic analyses (lines 225-233): Were the p-values for the Spearman correlations corrected for multiple testing?

·      Throughout the main body of the manuscript: Specifying whether they refer to the behavioral or neural PIT effect (at each mention) would greatly improve the clarity of the paper, especially in line 289 where the association with the (behavioral) PIT effect is mentioned in the same sentence immediately following the PIT-related amygdala activation.

·      Minor comments:

o  Line 39: Define CSs at first mention in the abstract

o  Line 73: Define first mention of fMRI 

o  Line 103: Insert space between major and neurological

o  Lines 231 and 232: Should be analysis

Author Response

The authors aimed to investigate non-drug-related Pavlovian-to-instrumental transfer (PIT) in a sample of young, male high- and low-risk social drinkers. They assessed behavioral PIT effects, BOLD activation in a priori defined regions of interest, namely, amygdala and nucleus accumbens, and associations with alcohol consumption, as well as a polygenic risk score for alcohol consumption. Results showed that behavioral PIT was increased in high-risk compared to low-risk drinkers. Across groups, there was a significant PIT-related BOLD activation in the right amygdala but not in the nucleus accumbens. There was a positive correlation between the behavioral PIT effect and the polygenic risk score for alcohol consumption. 

The present manuscript is well written. The rationale and methodology are clearly described, and the conclusions are justified by the data. Suggestions to improve the manuscript are listed below.

·      Abstract line 42: The authors should be precise with regard to which PIT effect they are referring to (behavioral or neural). 

We thank the reviewer for his/her suggestion and added behavioral and neural, respectively.

·      Abstract, line 43ff: The authors conclude that PIT-related amygdala activation might be a biomarker for a subclinical phenotype of risky alcohol consumption, however, such an association between PIT-related amygdala activation and the polygenic risk score or reported alcohol consumption is not reported in the manuscript. Was PIT-related amygdala BOLD activation associated with the polygenic risk score or alcohol consumption? If so, this should be reported; if not, this should still be reported, discussed, and reflected in the last sentence of the abstract.

We thank the reviewer for this important comment. PIT-related BOLD activation was not associated with PRS or alcohol consumption. We apologize for this misleading phrase in our abstract However, the behavioral PIT effect itself was, and this is now more clearly stated in lines (l. 46f).

The aim of the manuscript was to investigate, how behavioral PIT is related to i) risky alcohol consumption, ii) neural activation and iii) polygenic risk (we tried to make it clearer in the introduction, l. 58ff; and discussion, l. 299ff). The reviewer suggests very interesting additional analyses; however, with all due respect to this reviewer, we suggest to not include the analyses in the manuscript to avoid multiple testing. If the reviewer feels that these analyses need to be added, we will be happy to do so in the supplement.

·      Methods, 2.10. Polygenic analyses (lines 225-233): Were the p-values for the Spearman correlations corrected for multiple testing?

No, this was not corrected for multiple testing, as we conducted a replication (PRS and alcohol consumption). This replication is a proof of the PRS, whether it is related to alcohol consumption in our sample. We now state more clearly that we explored associations of this PRS with behavioral PIT, thus p-values are for descriptive reasons only (l. 248). To further explore associations between behavioral PIT, alcohol consumption, neural PIT and PRS, we conducted one model (see supplement S6).

·      Throughout the main body of the manuscript: Specifying whether they refer to the behavioral or neural PIT effect (at each mention) would greatly improve the clarity of the paper, especially in line 289 where the association with the (behavioral) PIT effect is mentioned in the same sentence immediately following the PIT-related amygdala activation.

We thank the reviewer for his suggestion to clarify the text body and accordingly added behavioral or neural PIT, respectively.

·      Minor comments:

o  Line 39: Define CSs at first mention in the abstract

o  Line 73: Define first mention of fMRI 

o  Line 103: Insert space between major and neurological

o  Lines 231 and 232: Should be analysis

Thank you for the minor comments. We accordingly change the respective phrases.

Reviewer 2 Report

Garbusow et al., present an interesting study on how pavlovian-instrumental transfer relates to different populations of drinkers and their genetic/neural sequele. A particular strength is the size of their data set and the robustness of the behavioral paradigm across subjects. This is evident in the relatively subtle difference in PIT effect between their drinking groups that is brought to statistical significance because of the combination of the behavioral platform and sample size. The ability of the behavioral effects to predict drinking phenotype replicates many studies in the field and the additional neural/genetic analysis potentially contribute unique findings. However, the methodology and analyses could be made clearer with some of the claims and rationale for the study being more carefully considered. These are detailed below.

--The rationale for using only male subjects is weak and not accurate. It is not true that male subjects have more problematic drinking. Male drinking tends to be more overt and the associated behaviors are more visible. A growing literature shows that female drinking is just as problematic and problematic drinking in females is in fact growing at a faster rate than males as are the manifestations of that problematic drinking. Female problematic drinking simply manifest in ways that lack some of the very public and overt ways that male drinking presents. Please remove that justification statement from the manuscript given that the depiction of male drinking as more problematic is not accurate.

--The use of “healthy social drinkers” to determine genetic risk (the training data set) is not compatible with the goal to investigate problem drinking. How could genes be identified as associated with heavy drinking if only social drinkers are used in the initial identification of genes?

--The message that the PIT effect is higher in the at-risk sample doesn’t accurately represent the results (which don’t show this, only a statistical correlation across all risk scores) or the sample (there isn’t a high-risk sample, as they further discuss the absence of actual heavy drinkers). The authors are overstretching their results here; there are no problematic or even “high risk” drinkers here because it is not clear whether their different populations are actually more or less at risk based on the current literature. They should be more descriptive with their analysis and label their groups according their measurable drinking behavior rather than trying to make some statement about risk.

--Fig 5B,D - I don’t understand the threshold application with the PRS. If the threshold is 1 then all SNPs are included, meaning even those not associated with alcohol intake; then you’d hope to see a larger proportion of the variance explained with the lower p value thresholds applied, where only alcohol-associated SNPs are considered. The opposite is seen in the data.

--The methods are lacking in many places to be sufficient to understand the study.

One place is the PIT procedure where it is very unclear how many CS are actually paired/unpaired with the coin, how those CS+/- are counterbalanced and what their response to the CS- vs. CS+ looks like. One thing of concern is that a CS with value of -2 in Figure 3 produces 4 button presses whereas one with a value of 2 produces  ~6 responses. That is not very good discrimination between the CS+ and the CS-. It seems their cohort may not have been trained to criterion properly?

For the imaging, when exactly was the imaging data collected relative the CS presentation and responses? If the amygdala p-values are corrected for multiple comparisons (e.g., NaCC, the other brain wide-measures), are they still significant?

            For the polygenic analysis, where did the genetic information come from for these             subjects, when was it collected, how, how were the SNPs determined (e.g., which             platform), etc? More information is needed on how these analy

Author Response

Garbusow et al., present an interesting study on how pavlovian-instrumental transfer relates to different populations of drinkers and their genetic/neural sequele. A particular strength is the size of their data set and the robustness of the behavioral paradigm across subjects. This is evident in the relatively subtle difference in PIT effect between their drinking groups that is brought to statistical significance because of the combination of the behavioral platform and sample size. The ability of the behavioral effects to predict drinking phenotype replicates many studies in the field and the additional neural/genetic analysis potentially contribute unique findings. However, the methodology and analyses could be made clearer with some of the claims and rationale for the study being more carefully considered. These are detailed below.

--The rationale for using only male subjects is weak and not accurate. It is not true that male subjects have more problematic drinking. Male drinking tends to be more overt and the associated behaviors are more visible. A growing literature shows that female drinking is just as problematic and problematic drinking in females is in fact growing at a faster rate than males as are the manifestations of that problematic drinking. Female problematic drinking simply manifest in ways that lack some of the very public and overt ways that male drinking presents. Please remove that justification statement from the manuscript given that the depiction of male drinking as more problematic is not accurate.

We thank the reviewer for his/her important comment and are fully in line with him/her, that problematic drinking is prominent and problematic in females, too. When planning the study, we were geared to Livingston et al. (2009). These authors reported higher problematic drinking among male subjects. We now added a statement in our limitation to underline the importance of investigating behavioral mechanisms as PIT in female problematic drinking, too (l. 359ff).

--The use of “healthy social drinkers” to determine genetic risk (the training data set) is not compatible with the goal to investigate problem drinking. How could genes be identified as associated with heavy drinking if only social drinkers are used in the initial identification of genes?

We agree with the reviewer that the optimal setup for a risk score analysis is using the exact same phenotype for training and target phenotype. On the other hand, the scoring precision relies heavily on the sample size of the training meta-analysis. Also it is clear that there is a lot of common genetics across related psychiatric phenotypes. So we decided to go for the largest training analysis possible while using a related psychiatric phenotype to achieve maximum power with the currently available data.

--The message that the PIT effect is higher in the at-risk sample doesn’t accurately represent the results (which don’t show this, only a statistical correlation across all risk scores) or the sample (there isn’t a high-risk sample, as they further discuss the absence of actual heavy drinkers). The authors are overstretching their results here; there are no problematic or even “high risk” drinkers here because it is not clear whether their different populations are actually more or less at risk based on the current literature. They should be more descriptive with their analysis and label their groups according their measurable drinking behavior rather than trying to make some statement about risk.

We thank the reviewer for this important comment. We didn`t want to overstretch our results. We used a definition of high versus low risky drinking (for acute alcohol related problems) according to WHO (2000, Reference 17 in our text). We introduced this definition by WHO very precisely in the methods part (see l. 106ff). While we feel that we should not change the label defined by the WHO, we agree with the reviewer that high intake does not necessarily predict a high risk is future consumption. We now state this in the Limitations (l. 352ff).

--Fig 5B,D - I don’t understand the threshold application with the PRS. If the threshold is 1 then all SNPs are included, meaning even those not associated with alcohol intake; then you’d hope to see a larger proportion of the variance explained with the lower p value thresholds applied, where only alcohol-associated SNPs are considered. The opposite is seen in the data.

We refer to the introduction of polygenic risk scoring (e.g. Purcell, Nature, 2009). In the threshold of 1 also the significant variants are included and naturally up-weighted due to their bigger effect-sizes. In fact, it is nowadays almost standard to use the most inclusive threshold if on concentrates on only one, especially in psychiatric phenotypes. In summary we appreciate the reviewer‘s concerns but would like to emphasize that our approach is currently the standard one.

--The methods are lacking in many places to be sufficient to understand the study.

One place is the PIT procedure where it is very unclear how many CS are actually paired/unpaired with the coin, how those CS+/- are counterbalanced and what their response to the CS- vs. CS+ looks like. One thing of concern is that a CS with value of -2 in Figure 3 produces 4 button presses whereas one with a value of 2 produces  ~6 responses. That is not very good discrimination between the CS+ and the CS-. It seems their cohort may not have been trained to criterion properly?

For the Pavlovian conditioning part (see figure 2B), we paired five CSs with pictures of a coin (US; -2€,-1€,0€,1€,2€). The CS-US association was counterbalanced between subjects, but deterministically within subjects. Each pairing was shown 16 times, resulting in 80 trials for the Pavlovian conditioning.

During the transfer part, we showed one out of five CS in the background only (no coins anymore) and in the foreground one out of six shells (asking the subjects to respond to the shells as learned in the instrumental part; but under nominal extinction, i.e. no reward shown after each trial; see figure 2C). The stimulus combination (CS in background and shell in foreground) was counterbalanced. Every combination is shown 3 times (randomized), resulting in 90 trials for the transfer phase (we added a phrase in the main text to clarify, l.154ff). When we compute the PIT effect, we collapse all instrumental responses during the transfer part, i.e. “collect” and “not collect” responses. This explains, why although CS- is in the background, we observe button presses. Crucially, CS- reduce the number of button presses, while CS+ enhances number of button presses (Figure 3). Please note that these button presses were performed to attain the goal in the instrumental responding part of the task. This instrumental part as now stated in line 152f of the task description, was unrelated to the Pavlovian background stimuli. Nevertheless, the Pavlovian background stimuli promoted approach in instrumental responses when they were positive and inhibited approach when they were negative. The observation that still about 4 button presses occurred when the aversive/negative Pavlovian stimuli appeared in the background is due to the fact that negative Pavlovian stimuli are combined at random with valuable targets in the instrumental condition, so instrumental approach is inhibited by Pavlovian cues even though they are irrelevant for the instrumental task. What we show in figure 3 is not instrumental performance per se, but the instrumental response modified by the respective background stimulus. The subjects properly learned the instrumental contingencies, as you can see in the supplementary results table S3 (number of button presses is significantly explained by instrumental contingency. i.e. higher number of button presses during “collect”; lower number of button presses during “not collect”. In the main text, we only refer to the PIT effect (modulation of instrumental response by background stimuli) as we aimed to focus on the PIT effect for clarity.  

For the imaging, when exactly was the imaging data collected relative the CS presentation and responses? If the amygdala p-values are corrected for multiple comparisons (e.g., NaCC, the other brain wide-measures), are they still significant?

The imaging data were collected during the Transfer part of the PIT paradigm, i.e. after the instrumental and Pavlovian conditioning. We first showed the Pavlovian background pictures for 600ms and then added the instrumental stimulus to the foreground (one out of six shells, respectively) for another 3 sec (the background stimulus was still visible). The response was conducted during these 3 sec. To control for motor activation in our PIT effect, we included a stick function for button press in our imaging first level model.

Our ROI encompassed the bilateral Nacc plus amygdala and thus was adequately corrected for multiple testing as by default implemented in SPM 12.

Within this ROI, we found a significant activation of the right amygdala during PIT (ROI used from WFU pick atlas); extracting individual mean values from the right amygdala and confirmed a direct correlation with the strength of the individual behavioral PIT effect (see Figure 4B). This is standard approach for ROI analyses.

            For the polygenic analysis, where did the genetic information come from for these             subjects, when was it collected, how, how were the SNPs determined (e.g., which             platform), etc? More information is needed on how these analy

The PRS was based on a GWAS in an independent (large) sample (from Schumann et al., PNAS, 2016, see reference 33 in our main text), in which the phenotype was problem drinking. So just using the weighting of the individual SNPs from that study, we calculated the PRS from the genotypes of the participants in our own sample. Although not all problem drinkers, part of the sample was. As such, these should also load higher on the GWAS derived PRS.

We added information about genotyping our sample in the methods part and apologize for not providing the information earlier (l. 185ff).

Unfortunately, this comment by the reviewer seems incomplete. Therefore, we hope, that we sufficiently answered to your comment. If not, we will be more than happy to do so in a second revision.

Round 2

Reviewer 2 Report

Garbusow et al., submitted an improved manuscript that was responsive to review. Many of the key concerns were thoughtfully addressed and explained. The study presents valuable data in two different populations of drinkers across behavioral, genetic and brain system. Remaining concerns are detailed below.

-I appreciate the author’s response to the comment about problem drinking in both sexes. However, the authors need to remove or precisely qualify the statement that “problematic drinking is higher in males compared to females” on page 3, line 114. That statement is simply not accurate given the broader literature outside the Livingston et al., 2009 paper they cite.  For example, see Keyes et al., ACER 2019 as a good example of how the definition of problematic drinking informs sex differences in drinking as does a more careful stratification of the population.

-I better understand their point about how they categorized alcohol drinking risk. But it is still strange that they use categorical terms like “high-“ and “low-“ risk drinkers but then perform largely correlational analyses (e.g., the imaging and the genetic data). For example, if they simply compare the activation of the amygdala to PIT in their high-risk group to their low-risk groups is there a difference? Right now they show a correlation but do not retain their high- and low-risk grouping in the analysis. The paper would be much improved if they were more careful to describe their findings on a spectrum rather than the categorical terms they are currently using unless they actually show a difference between the categorical high-/low-risk dataset across all metrics (behavioral, imaging and genetic analysis).

-The multiple comparisons are still not accurately accounted for in their imaging analysis. The authors effectively had 5 different brain regions they examined in their imaging data; the Amygdala, the Nacc, the R Pallidum, the R Superior Temporal Gyrus and the L Hippocampus. The multiple comparisons correction needs to be applied for 5 all regions (encompassing the experimental and control regions) rather than only the 2 experimental regions that are currently presented.

Author Response

Garbusow et al., submitted an improved manuscript that was responsive to review. Many of the key concerns were thoughtfully addressed and explained. The study presents valuable data in two different populations of drinkers across behavioral, genetic and brain system. Remaining concerns are detailed below.

-I appreciate the author’s response to the comment about problem drinking in both sexes. However, the authors need to remove or precisely qualify the statement that “problematic drinking is higher in males compared to females” on page 3, line 114. That statement is simply not accurate given the broader literature outside the Livingston et al., 2009 paper they cite.  For example, see Keyes et al., ACER 2019 as a good example of how the definition of problematic drinking informs sex differences in drinking as does a more careful stratification of the population.

We thank the reviewer for the comment on gender differences. We agree, that gender differences in alcohol related problems are more complex than we illustrate in our manuscript. We now removed the sentence in our methods (l. 107f) and discussion (l. 365f) section.

-I better understand their point about how they categorized alcohol drinking risk. But it is still strange that they use categorical terms like “high-“ and “low-“ risk drinkers but then perform largely correlational analyses (e.g., the imaging and the genetic data). For example, if they simply compare the activation of the amygdala to PIT in their high-risk group to their low-risk groups is there a difference? Right now they show a correlation but do not retain their high- and low-risk grouping in the analysis. The paper would be much improved if they were more careful to describe their findings on a spectrum rather than the categorical terms they are currently using unless they actually show a difference between the categorical high-/low-risk dataset across all metrics (behavioral, imaging and genetic analysis).

We feel happy about the clarification why we use the terms “high-“ and “low-risk drinkers”. The correlational analyses we showed, were for validation of our data. We indeed found no differences between groups regarding the neural PIT effect, which is now more clearly stated in the Results section (l. 277f). Second, the correlation between PRS and alcohol consumption shows that the PRS computed with our data is related to alcohol consumption as it was the case in the Schumann cohort. This legitimates to use the PRS in our sample for further analyses such as the association with the behavioral PIT.

-The multiple comparisons are still not accurately accounted for in their imaging analysis. The authors effectively had 5 different brain regions they examined in their imaging data; the Amygdala, the Nacc, the R Pallidum, the R Superior Temporal Gyrus and the L Hippocampus. The multiple comparisons correction needs to be applied for 5 all regions (encompassing the experimental and control regions) rather than only the 2 experimental regions that are currently presented.

With all due respect to the reviewer, we disagree, because we did not - as she or he feels - use five different ROIs but instead we used one ROI which was adequately corrected voxelwise for multiple testing. As stated in the introduction (l. 83ff) and the methods section (l. 236ff), we a priori defined one ROI that includes limbic structures such as the amygdala and nucleus accumbens, regions typically activated by PIT. For this ROI, we accounted properly for multiple comparisons (small volume correction; FWE p<0.05). In one single ROI you do not correct for multiple testing a second time regarding different brain regions included in the ROI. We then extracted the individual values of the mean activation of this ROI and showed that these individual mean values also correlate significant with the behavioral PIT effect. This is a standard approach for ROI analyses (e.g. Poldrack, 2007; Lessov-Schlaggar et al., 2013; Schacht et al., 2013; Garbusow et al., 2016; Seo et al., 2018).

We also present an analysis in the supplement that explored brain areas outside of the predefined and hypothesis-based ROI showing the regions activated by PIT in the whole brain. Note, that this exploratory analysis is uncorrected for multiple comparisons and thus presented for descriptive reasons only. It is not acceptable to use these regions for further analyses, also not applying Post hoc corrections for multiple regions.

References:

Garbusow M, Schad DJ, Sebold M, Friedel E, Bernhardt N, Koch SP, Steinacher B, Kathmann N, Geurts DE, Sommer C, Müller DK, Nebe S, Paul S, Wittchen HU,

Zimmermann US, Walter H, Smolka MN, Sterzer P, Rapp MA, Huys QJ, Schlagenhauf F, Heinz A. Pavlovian-to-instrumental transfer effects in the nucleus accumbens relate to relapse in alcohol dependence. Addict Biol. 2016;21(3):719-31. doi:10.1111/adb.12243.

Lessov-Schlaggar CN, Lepore RL, Kristjansson SD, Schlaggar BL, Barnes KA,Petersen SE, Madden PA, Heath AC, Barch DM. Functional neuroimaging study inidentical twin pairs discordant for regular cigarette smoking. Addict Biol. 2013 Jan;18(1):98-108. doi: 10.1111/j.1369-1600.2012.00435.x.

Poldrack RA. Region of interest analysis for fMRI. Soc Cogn Affect Neurosci. 2007;2(1):67–70. doi:10.1093/scan/nsm006.

Schacht JP, Anton RF, Myrick H. Functional neuroimaging studies of alcohol cue reactivity: a quantitative meta-analysis and systematic review. Addict Biol. 2013;18(1):121–133. doi:10.1111/j.1369-1600.2012.00464.x.

Seo J, Moore KN, Gazecki S, et al. Delayed fear extinction in individuals with insomnia disorder. Sleep. 2018;41(8):zsy095. doi:10.1093/sleep/zsy095.